# Effect of Vitamin E on Diabetic Nephropathy in Streptozotocin-Induced Diabetic Rats

**DOI:** 10.3390/ijms26041597

**Published:** 2025-02-13

**Authors:** David Segura Cobos, Esperanza Enedina Díaz Salgado, Dante Amato, Sinaí Ernesto Cardoso García, Tomás Ernesto Villamar Duque, Anayantzin Paulina Heredia Antúnez, Leonardo del Valle Mondragón, Gil Alfonso Magos Guerrero, Elizabeth Alejandrina Guzmán Hernández

**Affiliations:** 1Faculty of Higher Studies Iztacala, National Autonomous University of Mexico (UNAM), Tlalnepantla 54090, Mexico; seguracd@unam.mx (D.S.C.); espe_ash@hotmail.com (E.E.D.S.); dante.amato@unam.mx (D.A.); 2Bioterium, Universidad Panamericana, Mexico City 03920, Mexico; cardosogarcia@outlook.com; 3General Bioterium, Faculty of Superior Studies Iztacala, Biology, National Autonomous University of Mexico (UNAM), Tlalnepantla 54090, Mexico; vidutoer@yahoo.com.mx (T.E.V.D.); paulina_852@hotmail.com (A.P.H.A.); 4Department of Pharmacology, National Institute of Cardiology Ignacio Chavez, Mexico City 04510, Mexico; leonardodvm65@hotmail.com; 5Department of Pharmacology, Faculty of Medicine, National Autonomous University of Mexico (UNAM), Mexico City 04510, Mexico; gamagos@unam.mx

**Keywords:** diabetic nephropathy, vitamin E, AT1 receptor, antioxidant

## Abstract

Diabetic nephropathy (DN) is a serious complication of diabetes mellitus; oxidative stress plays a key role in the pathogenesis of DN. The objective of this study was to evaluate the antioxidant effect of vitamin E on diabetic nephropathy. A control group and three groups of rats with streptozotocin-induced diabetes mellitus (untreated diabetic rats and diabetic rats treated with vitamin E 250 and 500 mg/kg) were studied. After 4 weeks of treatment, the kidneys were removed under anesthesia with sodium pentobarbital. The kidneys were weighed, the AT_1_ and AT_2_ receptor expression was measured by Western blot, and the activities of glutathione peroxidase, catalase, and superoxide dismutase were determined in the renal cortex. Rats with diabetes mellitus had hyperglycemia, increased food and water consumption, and higher urinary volume than control rats. In diabetic rats (DM), kidney hypertrophy was observed and measured by kidney weight, protein/DNA ratio in the renal cortex, and proximal tubular cell area; proteinuria and reduced creatinine clearance were observed. AT_1_ and AT_2_ receptor expression in the kidney cortex of DM rats increased significantly compared to normoglycemic rats; antioxidant enzyme activities were decreased; treatment with vitamin E reversed kidney hypertrophy and reduced proteinuria; reduction in expression of AT_1_ and AT_2_ receptors was associated with increased antioxidant activity. Thus, treatment with vitamin E slows the progress of DN.

## 1. Introduction

Diabetes mellitus has become a public health problem. It is estimated that around 463 million people between the ages of 20 and 79 years have this condition, and this figure is expected to reach 643 million by 2030 [1]. According to the National Survey of Public Health and Nutrition in Mexico (ENSANUT 2021), there are 12.4 million people with diabetes mellitus [2]. Multiple causes can lead to a person developing this disease, including following an unhealthy lifestyle. This includes an increase in the consumption of ultra-processed foods, excessive consumption of sugary drinks, and deficiency in consumption of fruits, vegetables, and fiber. In addition, the vast majority of the population does only a few hours of exercise, with many following a highly sedentary lifestyle [3]. Diabetes mellitus can cause long-term complications, which are divided into microvascular and macrovascular. The most relevant for the present study is microvascular conditions, in which capillary diffusion is affected in the target organs, mainly the kidney. Diabetic nephropathy is a complication that affects 15 to 25% of patients with type 1 diabetes mellitus and between 30 and 40% of type 2 diabetic patients. The vast majority of patients present renal hypertrophy, hyperfiltration, microalbuminuria, macroalbuminuria, and arterial hypertension, along with a progressive decrease in the glomerular filtration rate, the clinical evolution of diabetic nephropathy, up to renal failure [4]. Various mechanisms are involved in the development of diabetic nephropathy, among which the renin–angiotensin–aldosterone system has been identified as one of the most important. Several cellular events are activated during diabetic nephropathy, such as excessive channeling of glucose intermediaries into metabolic pathways with generation of advanced glycation products, activation of protein kinase C, increased expression of profibrotic cytokine transforming growth factor β and GTP-binding proteins, and production of reactive oxygen species. Hyperglycemia stimulates the synthesis and intrarenal secretion of angiotensin II, which causes constriction of mesangial cells and afferent arterioles when it binds to its AT1 receptor, leading to glomerular hypertension, which appears in the early stages of diabetic nephropathy. Angiotensin II acts as a growth factor for renal cells, inducing the expression and renal synthesis of autocrine factors and cytokines, such as transforming growth factor beta, which is activated through the generation of oxidative stress. Angiotensin II activates nicotinamide adenine phosphate dehydrogenase (NADPH) through its subunit NOX4, which is found in the kidney, and can increase the synthesis of TGF-b1, which acts by stimulating the production of extracellular matrix by mesangial and epithelial cells and fibroblasts. This leads to the development of fibrosis, causing kidney damage until it culminates in the need to replace this organ [5]. There are few treatment options for patients with kidney failure and decreased glomerular filtration rate. Usually, treatment is based on insulin; however, recent studies have shown that the use of non-enzymatic antioxidants, such as vitamins C, D, E, and b-carotene, helps reduce the generation of reactive oxygen species and increase endogenous antioxidant activity [6]. The objective of this study was to evaluate the antioxidant effect of vitamin E on diabetic nephropathy.

## 2. Results

Table 1 shows whole-animal data for control, diabetes mellitus (DM), and diabetes mellitus + vitamin E 250 and 500 mg/kg rats, including glycemia, body weight, consumption of water, and urinary volume. All STZ-induced diabetic rat groups had significant hyperglycemia (plasma glucose: DM, 406 ± 19 mg/dL vs. normoglycemic control 127 ± 7 mg/dL, *p* < 0.05) at the time of the experiment. In the diabetes mellitus and vitamin E 250 and 500 mg/kg groups, glycemia did not differ from the DM group. In DM rats, body weight was lower than in the control group, and water ingestion and urinary volume were markedly increased in the three groups of diabetic animals.

### 2.1. Effect of Vitamin E on the Cellular Area of the Proximal Tubules

Representative photomicrographs (magnification ×40) of kidney cortex sections stained with HE from all experimental groups are shown in Figure 1a. The proximal tubule cellular area (Figure 1b) in DM rats increased significantly as compared to the normoglycemic control group (C: 68.81 ± 1.7, DM: 84.7 ± 1 μm^2^). Oral treatment with Vit E 500 mg/kg partially prevented this increment (DM + Vit E: 69.4 ± 0.8 μm^2^) (Figure 1b). 

### 2.2. Effect of Vitamin E on Renal Hypertrophy and Protein/DNA Ratio

In DM rats, the kidney weight/body weight ratio was significantly increased compared to the control normoglycemic group (C: 0.319 ± 0.02, DM: 0.487 ± 0.02 g) (Figure 2a); oral treatment with Vit E 250 and 500 mg/kg prevented this increment (0.413 ± 0.18 g and 0.416 ± 0.27 g). Also, the protein/DNA ratio was higher in DM than in C (C: 0.20 ± 0.003, DM: 0.30 ± 0.01 in arbitrary units); treatment with Vit E 250 and 500 mg/kg reversed this increment (0.18 ± 0.01 and 0.20 ± 0.02 in arbitrary units) (Figure 2b).

### 2.3. Effect of Vitamin E on Kidney Function

As an indicator of kidney function, the creatinine clearance was quantified. In DM rats, creatinine clearance was significantly lower compared to the control normoglycemic group (C: 0.22 ± 0.05, DM: 0.086 ± 0.01 mL/min); treatment with vitamin E 250 and 500 mg/kg prevents deterioration of creatinine clearance (Figure 3a). This indicator was matched by proteinuria being significantly increased in DM rats compared to the control normoglycemic group (C: 24 ± 3, DM: 35 ± 4 mL/24 h); oral treatment with vitamin E 250 and 500 mg/kg reversed this increment (DM + Vit E: 24 ± 3 and 11 ± 3 mL/24 h, Figure 3b). This result suggests an antihypertrophic effect of vitamin E.

### 2.4. Effect of Vitamin E on the Activities of Antioxidant Enzymes

The antioxidant enzymes superoxide dismutase, catalase, and glutathione peroxidase showed reduced activities in DM rats compared to the control normoglycemic rats (Figure 4a–c); vitamin E 250 and 500 mg/kg treatment restored the activities of antioxidant enzymes (Figure 4a–c).

### 2.5. Effect of Vitamin E on AT_1_ and AT_2_ Receptor Protein Expression in the Renal Cortex

AT_1_R protein expression in the renal cortex of DM rats increased significantly compared to the control normoglycemic group (Figure 5a). Treatment with oral vitamin E 250 and 500 mg/kg reversed AT_1_R protein overexpression (Figure 5a). AT_2_R protein expression in the renal cortex of diabetic rats increased significantly compared to the control group (Figure 5b).

## 3. Discussion

There are several models of DM induction in animals; the most reliable method, and therefore the most commonly used, is intraperitoneal administration of streptozotocin (STZ). This study evaluated the protective role of vitamin E in STZ-induced renal damage in rats as evidenced by histomorphological observations after treatment with doses of vitamin E 250 and 500 mg/kg for 4 weeks.

STZ is a nitrosourea that contains a glucose-like structural component, so it recognizes GLUT2 glucose transporters, causes decreased expression of these, and destroys a significant number of pancreatic beta cells by direct toxicity in DNA in a short time of only 24 h, thus triggering a state like DM. Intraperitoneal injection of STZ (60 mg/kg) into adult Wistar rats has been shown to induce DM within the first 2 to 4 days [7]. In the DM + Vit E 250 and 500 mg/kg groups, glycemia did not differ from the DM rats, identical to what was observed by other authors [8], who have suggested that mechanisms of vitamin E in improving control of glycemia include protecting islet b-cells by reducing the cytotoxicity mediated by cytokines and their products and possibly enhancing insulin action [9]. Vitamin E antioxidant activity affects the polyol pathway [9]. Vitamin E has systemic and renal beneficial effects, which are produced through its action on inflammation, vascular health, and oxidative stress [6]. The improvement of DN caused by vitamin E supplementation in rat models is mediated by activation of diacylglycerol kinase, which prevents abnormal activation of protein kinase C and podocyte degeneration by reducing the circulating level of diacylglycerol [10].

Polyuria is caused by excess glucose filtered in kidneys that are not reabsorbed in renal tubules; therefore, it is excreted in the urine. Non-reabsorbed glucose generates water retention by osmosis in the proximal contoured tubule, eliminating a higher volume of water and electrolytes. The results showed that DM rats excreted a higher urine volume than the control group. At the same time, polydipsia is a compensatory mechanism generated against the volume of liquid lost, increasing water excretion [11]. Several studies have shown that in the early stages of the development of DN, there is an increase in the weight of the kidney as well as the area of proximal tubular cells. Structural alteration is known as renal hypertrophy [12]; in DM, it has been shown that oxidative stress and free oxygen radicals, which develop during ND, trigger apoptosis of tubular epithelial cells and podocytes. Treatment with vitamin E plays a central role in the antioxidant protective system, protecting all lipids undergoing oxidation and diminishing the number of apoptotic cells [13]. In this study, renal hypertrophy was determined through the renal weight/total body weight ratio and was associated with the presence of proteins in urine; these are two indicators that allow us to infer renal damage caused by the breakdown of the glomerular filtration barrier due to a persistent hyperglycemic state, as this barrier prevents the passage of macromolecules such as plasma proteins by their size, shape, and negative electrical load [14].

As an indicator of kidney function, creatinine clearance in DM rats was significantly lower compared to the control group. With increased urinary protein levels associated with systemic and metabolic disorders such as DM, the kidney may undergo slow but progressive deterioration, eventually leading to renal failure. Renal involvement in these disorders is often first manifested by a gradual increase in proteinuria [15]. Treatment with vitamin E 250 and 500 mg/kg reversed this increment. This result suggests an antihypertrophic effect of vitamin E; in line with our results, it has been recently reported that vitamin E reduced blood urea nitrogen and creatinine [16].

Several studies have demonstrated that, under hyperglycemic conditions, the overproduction of free radicals, mainly ROS, is a major factor contributing to the progression of DM. The SOD, catalase, and GPx activities were reduced in DM compared to the control group, indicating the development of oxidative stress (ROS). In addition, ROS are also generated due to the interaction of glucose with protein, resulting in advanced glycation end products (AGEs), which block the receptors and inactivate the enzymes [17]. As AGEs have been found to be involved in the progression of many irreversible complications such as diabetes, the number of AGEs is also considered an important marker to estimate during nephrotoxicity [18]. In a study where the effect of vitamin E on the nephrotoxicity of diclofenac was studied (30 mg/kg, orally, 10 days), it was observed that vitamin E caused a significant reduction in KIM-1, TNF-α, and NF-κB, which increased due to the effect of diclofenac [19].

The results of the current study showed that vitamin E 250 and 500 mg/kg treatment restored activities of antioxidant enzymes, probably due to increased Nrf-2 transcription factor expression and inhibition of the chain reactions of lipid peroxidation [20], suggesting a possible renoprotective effect.

Another mechanism that participates in DN is the angiotensin renin system, which can be activated by hyperglycemia, leading to local release of angiotensin II, which affects glomerular cells and induces sclerosis dependent on their hemodynamic actions. Our findings agree with other authors who have suggested that Ang II, through its binding to the AT1 receptor, contributes to ND by activation of NADPH oxidase and generates reactive oxygen species [21]; treatment with vitamin E 250 and 500 mg/kg reversed AT_1_R protein expression. AT_2_R protein expression in the renal cortex of diabetic rats increased significantly compared to the control group, as has been shown by other authors [21], who additionally observed reduced tubular ACE2 and Ang- (1–7) expressions and a compensatory increase in AT_2_R expression in DM rats. Concomitant AT_2_R agonist and ACE2 activation significantly attenuated oxidative stress in DM rats.

## 4. Materials and Methods

We used the streptozotocin (STZ)-induced diabetes model. Male Wistar rats were obtained from the Bioterium of Facultad de Estudios Superiores Iztacala, UNAM. Animals aged 10 weeks with the initial body weight of 180 ± 20 g were studied. Rats had free access to standard rat chow (Rodent Laboratory Chow 5001, Ralston Purina, Richmond, IN, USA) and tap water, with 12 h light/12 h dark cycles during the experiment. Diabetes mellitus (DM) was induced by a single STZ intraperitoneal (ip) injection (60 mg/kg of body weight) dissolved in the vehicle 10 mM sodium citrate buffer with pH 4.5. Control (C) rats received vehicles alone. Two days after STZ injection, glycemia was determined in tail vein blood samples using a reflectance meter (One Touch; LifeScan, Milpitas, CA, USA) [13]. Only rats with blood glucose levels > 300 mg/dL were included in the experiments. Diabetic rats were randomized into 3 groups: (1) untreated diabetic rats (DM) receiving vehicle (oil), (2) diabetic rats treated with vitamin E 250 mg/kg, and (3) diabetic rats treated with vitamin E 500 mg/kg (DM + Vit E) [7]. Once hyperglycemia with serum glucose levels of ≥300 mg/dl was confirmed, treatment with vitamin E was given daily during the 4 weeks of the experiment. Each group consisted of 5 rats. Two days before the end of the experiment, animals were placed in metabolic cages to measure food and water consumption and urinary volume as well as obtain urine samples to measure proteins and creatinine.

At the end of the experiment, the rats were anesthetized with sodium pentobarbital (45 mg/kg, via intraperitoneal). Blood samples were obtained to measure glycemia. Both kidneys were quickly removed. The left kidney was decapsulated, weighed, and dissected into the cortex to determine the activity of antioxidant enzymes (superoxide dismutase, catalase, and glutathione peroxidase).

### 4.1. Measurement of Renal Hypertrophy

The kidney weight/rat body weight ratio was calculated as a kidney hypertrophy index. The formaldehyde-fixed right kidneys were dehydrated through an ethanol-graded series, embedded in paraffin, sectioned into 4 mm thick slices, mounted on glass slides, and stained with hematoxylin–eosin (HE). Glomeruli and proximal tubular cells were visualized using an optic microscope, and the areas were measured using a computer program (Motic Image Plus 2.0 ML, Richmond, BC, Canada). Next, 50 consecutive glomeruli per rat were analyzed with 10× magnification for the glomerular area. For the proximal tubular cells area, 100 cells per field at 40× magnification were counted, and 10 fields per slide were analyzed and averaged. Finally, total DNA and protein from cortex tissue were extracted and quantified by the Trizol reagent method (Invitrogen, Grand Island, New York, NY, USA); the protein/DNA ratio was used to index relative hypertrophy.

### 4.2. Analytical Methods

Proteinuria was determined by the Bradford method (with Coomassie brilliant blue G-250 dye, Bio-Rad Laboratories, Inc., Hercules, CA, USA), we used bovine serum albumin (BSA) (Sigma Chemical Co. St. Louis, MO, USA) as standard, and creatinine was determined with Cayman reagents (Cayman Chemical, Ann Arbor, MI, USA). The protein/creatinine urinary excretion ratio was calculated.

### 4.3. Western Blotting

Renal tissue was homogenized in 100 mM Tris (hydroxymethyl-aminomethane-tris-hydrochloride, Sigma, St. Louis, MO, USA), pH 7.4, incubated with a protease-inhibitor cocktail (Mini Complete EDTA-free protease inhibitor cocktail, Roche, Germany), and centrifuged at 10,000× *g* for 10 min to remove insoluble debris. The protein concentration of the supernatant was quantified using the Bradford method. Under reducing conditions, 50 mg of protein was loaded into a 10% SDS-PAGE mini-gel, and the separated proteins were transferred to polyvinylidene difluoride (PVDF) membranes (Amersham Hybond ECL, GE Health Care, Buckinghamshire, UK). The membranes were blocked with 5% nonfat milk in Tris-buffered saline (TBS) (pH 7.6) containing 0.05% Tween 20 (TBST) for 2 h at room temperature. Membranes were incubated for 16 h at 4 °C with a 1:1000 dilution of a rabbit polyclonal antibody to AT_1_R and AT_2_R (Santa Cruz Biotechnology Inc., Santa Cruz, CA, USA). After incubation with the primary antibody, the membranes were washed with TBST buffer and incubated with a 1:1000 dilution of horseradish-peroxidase-labeled goat anti-rabbit IgG secondary antibody (Zymed, Invitrogen, Grand Island, New York, NY, USA) at room temperature for 2 h. Visualization was performed with an enhanced chemiluminescence (ECL) Western blotting kit (Luminol, Santa Cruz Biotechnology Inc., Santa Cruz, CA, USA). The obtained films were scanned and digitized using a flatbed scanner. Using the software Multi Gauge, Fuji Film Science, Lab2003 (Fuji Photo Film Co., Ltd., Tokyo, Japan), densitometry was measured by computer analysis. All membranes were stripped, re-blocked, and incubated with goat β-actin antibody (Santa Cruz Biotechnology Inc., Santa Cruz, CA, USA) as a housekeeping protein with the same protocol [13].

### 4.4. Evaluation of Oxidative Stress

Renal catalase (CAT) activity was assayed at 25 °C; the method was based on the disappearance of H_2_O_2_ from a solution containing 30 mmol/L H_2_O_2_ in 10 mmol/L potassium phosphate buffer (pH 7) at 240 nm. Glutathione peroxidase (GPx) activity was assessed by a method previously described; results were expressed as UI/mg protein. Superoxide dismutase (SOD) activity in kidney cortical homogenate was measured by a competitive inhibition assay using xanthine–xanthine oxidase system to reduce nitro blue tetrazolium [22].

### 4.5. Statistical Analysis

All values are presented as mean ± standard error of the mean (SEM) and compared by ANOVA followed by Newman–Keuls test. Differences were considered statistically significant with *p* < 0.05.

## 5. Conclusions

Treatment with vitamin E in diabetic rats prevents the development of proteinuria and renal hypertrophy, normalizes kidney function and antioxidant enzyme activities, and prevents overexpression of receptors for angiotensin II. Thus, treatment with vitamin E slows the progress of ND.

## Figures and Tables

**Figure 1 ijms-26-01597-f001:**
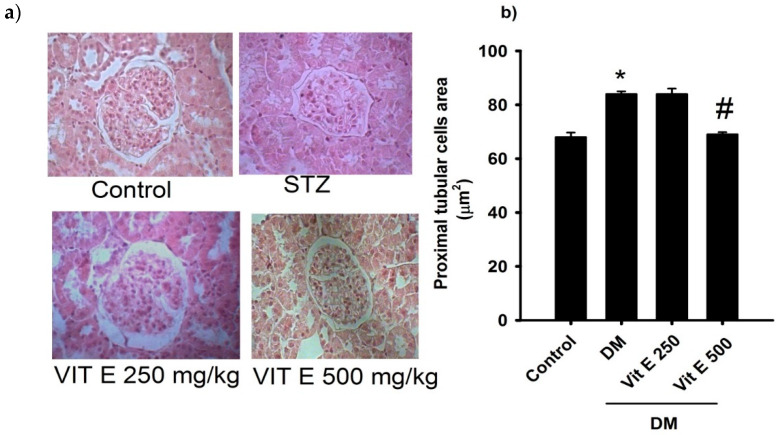
(**a**) Representative photomicrographs (magnification ×40) of kidney cortex sections stained with HE for the following groups: control normoglycemic, untreated diabetes mellitus (DM), diabetes mellitus treated with Vit E 250 (DM + Vit E 250 mg/kg), and diabetes mellitus treated with vitamin E 500 (DM + Vit E 500 mg/kg). (**b**) Comparison of proximal tubular cell area in the rats after 4 weeks of treatment. Data are expressed as mean ± SEM. * *p* < 0.05 vs. C, and # *p* < 0.05 vs. DM. Units = mm^2^.

**Figure 2 ijms-26-01597-f002:**
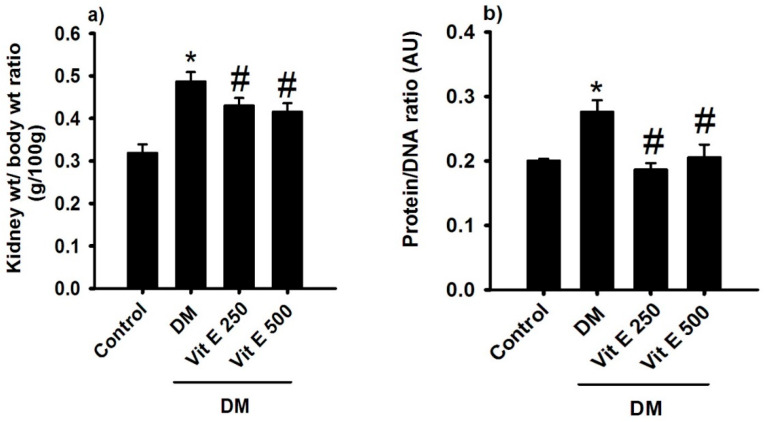
(**a**) Kidney weight/body weight ratio and (**b**) protein/DNA ratio for the following groups: control, untreated diabetes mellitus (DM), diabetes mellitus treated with Vit E 250 (DM + Vit E 250 mg/kg), and diabetes mellitus treated with vitamin E 500 (DM + Vit E 500 mg/kg). Data are expressed as mean ± SEM. * *p* < 0.05 vs. C, and # *p* < 0.05 vs. DM. AU is arbitrary units.

**Figure 3 ijms-26-01597-f003:**
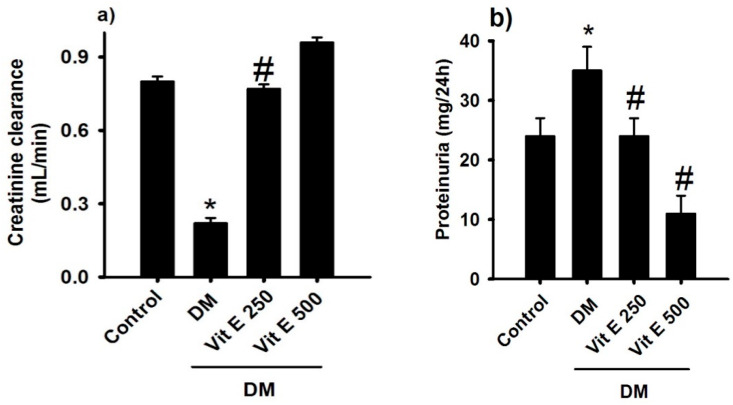
Comparison of the (**a**) creatinine clearance and (**b**) proteinuria for the following groups: control normoglycemic (C), untreated diabetes mellitus (DM), diabetes mellitus treated with vitamin E 250 (DM + Vit E 250 mg/kg), and diabetes mellitus treated with vitamin E 500 (DM + Vit E 500 mg/kg). Data are expressed as the mean ± SEM. * *p* < 0.05 C, # *p* < 0.05 vs. DM.

**Figure 4 ijms-26-01597-f004:**
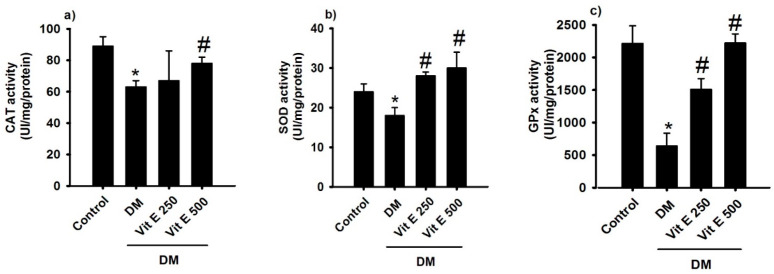
Comparison of antioxidant enzymes: (**a**) catalase (CAT), (**b**) superoxide dismutase (SOD), and (**c**) glutathione peroxidase activities (GPx) for the following groups: control, untreated diabetes mellitus (DM), diabetes mellitus treated with oral vitamin E 250 (DM + Vit E 250 mg/kg), and diabetes mellitus treated with oral vitamin E 500 (DM + Vit E 500 mg/kg). Data are expressed as mean ± SEM. * *p* < 0.05 C, # *p* < 0.05 vs. DM.

**Figure 5 ijms-26-01597-f005:**
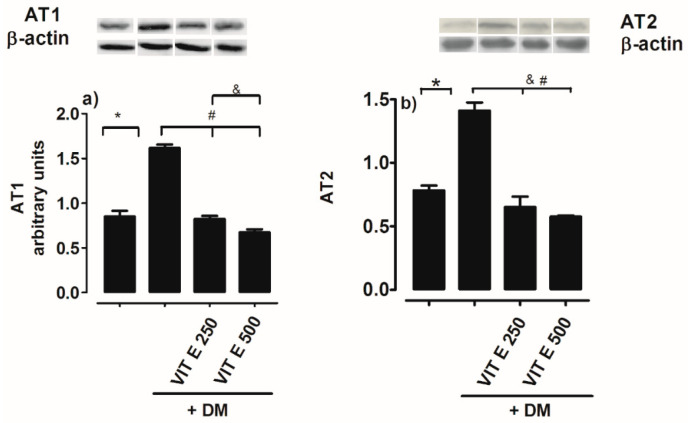
(**a**) AT_1_R and (**b**) AT_2_R protein expression in renal cortex homogenates for the following groups: control normoglycemic rats (C), untreated diabetes mellitus (DM), diabetes mellitus treated with vitamin E 250 mg/kg (DM VIT E 250), and diabetes mellitus treated with vitamin E 500 mg/kg (DM + VIT E 500). Data are expressed as mean ± SEM. * *p* < 0.05 C, # *p* < 0.05 vs. DM, and & *p* < 0.05 vs. treatments.

**Table 1 ijms-26-01597-t001:** Comparison of glycemia, body weight, water, food ingestion, and urinary volume in four groups of rats: control (c), untreated diabetes mellitus (DM), diabetes mellitus treated with vitamin E 250 (DM + Vit E 250 mg/kg), and diabetes mellitus treated with vitamin E 500 (DM + Vit E 500 mg/kg). Data are expressed as mean ± SEM. * *p* < 0.05 vs. Control.

Parameter	Control	DM	DM + Vit E 250 mg/kg	DM + Vit E 500 mg/kg
Glycemia (mg/dL)	127 ± 7	406 ± 19 *	405 ± 18	405 ± 23
Body weight (g)	341 ± 10	214 ± 24 *	246 ± 14 *	263 ± 8 *
Water ingestion (mL/24 h)	81 ± 14	161 ± 2 *	169 ± 10 *	168 ± 10 *
Urinary volume (mL/24 h)	10 ± 7	90 ± 14 *	124 ± 14 *	120 ± 10 *

## Data Availability

The original contributions presented in this study are included in the article. Further inquiries can be directed to the corresponding author.

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
