# Peer review of "Effect of Vitamin E on Diabetic Nephropathy in Streptozotocin-Induced Diabetic Rats"

_ijms, 2025, doi:10.3390/ijms26041597_

Round 1

Reviewer 1 Report

Comments and Suggestions for Authors

Comments and suggestions will be indicated the world document.

Author Response

We greatly appreciate your comments on our writing and in our next experimental processes we will consider your suggestion of increasing the number of rats, also taking into account the instructions given to us by the Ethics Committee of our institution.

Reviewer 2 Report

Comments and Suggestions for Authors

‘Effect of vitamin E on diabetic nephropathy in streptozotocin induced diabetic rats.’ might be beneficial for future studies. However, the quality of the data without RNA experiment is too poor to discuss the preventive effect of vitamin E and molecular mechanism of the dietary effect. Because there are no major remarks to the scientific content or the general message of the paper, I do not think they provide the necessary level for publication in IJMS.

1. The number of rat in the group is too low. They should increase the number of rat in STZ group and STZ/vitE group.

2. If the fasting blood glucose level remained unaltered with the vitE treatment compared to the DN group, then how does the protective effect of the vitE work? Because renal tubular epithelial cells are responsible for urinary glucose reabsorption, renal tubulointerstitial injury is an important pathological basis for DN progressing to end-stage renal diseases. They should examine RNA expression related to proximal tubule injury markers (Kim1, Krt20, Vcam1, Ccl2, and Cd44) to suggest that vitE supplementation protects the proximal tubules. Also, they should check fibrosis marker genes expression (fibronectin 1, transforming growth factor beta and Collagen1a1.

3. They showed AT1R and AT2R protein expression in renal cortex homogenate, but they should show the data from one photo, because they combined the different photos to compare the protein expression level.

Author Response

1.The number of rat in the group is too low. They should increase the number of rats in the STZ and STZ/vitE groups.

The number of rats in the different experimental groups has regularly been 5-6, during the different investigations that we have carried out in our pharmacology laboratory (FES Iztacala, UNAM). We could consider increasing the number of rats per experimental group from now on, based on your kind suggestion, also in accordance with the guidelines of the Faculty's Bioethics Commission.

  1. If the fasting blood glucose level remained unaltered with the vitE treatment compared to the DN group, then how does the protective effect of the vitE work? Because renal tubular epithelial cells are responsible for urinary glucose reabsorption, renal tubulointerstitial injury is an important pathological basis for DN progressing to end-stage renal diseases. They should examine RNA expression related to proximal tubule injury markers (Kim1, Krt20, Vcam1, Ccl2, and Cd44) to suggest that vitE supplementation protects the proximal tubules. Also, they should check fibrosis marker genes expression (fibronectin 1, transforming growth factor beta and Collagen1a1.

We did not observe a hypoglycemic effect of vitamin E at doses of 250 and 500 mg/kg in the rat diabetes mellitus model induced by intraperitoneal administration of streptozotocin 65 mg/kg, as well as other researchers such as Reno-Bernstein et al., 2022, they also showed. This model represents type 1 diabetes mellitus, where the recommended hypoglycemic treatment in people is only parenteral insulin.

Reno-Bernstein CM, Oxspring M, Bayles J, Huang EY, Holiday I, Fisher SJ. Vitamin E treatment in insulin-deficient diabetic rats reduces cardiac arrhythmias and mortality during severe hypoglycemia. Am J Physiol Endocrinol Metab. 2022 Nov 1;323(5):E428-E434. doi: 10.1152/ajpendo.00188.2022. Epub 2022 Oct 5. PMID: 36198111; PMCID: PMC9639754.

The renoprotective effect of vitamin E occurs due to its antioxidant properties that involve stimulating the activity of antioxidant enzymes such as catalase, superoxide dismutase and glutathione peroxidase,  in diabetic rats by induction with streptozotocin, probably through activation of the transcription factor Nrf2. Vitamin E also prevents the overexpression of the vasoactive peptide angiotensin II that induces renal hypertrophy through inducing the expression of cytokines such as transforming growth factor beta 1. The improvement of DN caused by vitamin E supplementation in rat models is mediated by activating diacylglycerol kinase, which prevents abnormal activation of protein kinase C and podocyte degeneration by reducing the circulating level of diacylglycerol [Jin et al., 2024]. 

Jin. Z.; Sun. J.; Zhang. W. Effect of Vitamin E on Diabetic Nephropathy: A Meta-Analysis. Altern Ther Health Med. 2024;30(9):344-349.

Treatment with vitamin E plays a central role in the antioxidant protective system, protecting all lipids undergoing oxidation and diminishing the number of apoptotic cells [Amato et al., 2016]. Vitamin E reduces oxidative stress due to its antioxidant property and has positive effects on renal function parameters, thus providing a protective effect in DN (Zhao et al., 2018).

Dante Amato.; Alma. R. Núñez-Ortiz.; José del Carmen. Benítez-Flores.; David. Segura-Cobos.; Pedro. López-Sánchez.; Beatriz. Vázquez-Cruz. Role of Angiotensin-(1-7) on Renal Hypertrophy in Streptozotocin-Induced Diabetes Mellitus. Pharmacology & Pharmac. 2016;379-395.

Zhao Y, Zhang W, Jia Q, Feng Z, Guo J, Han X, et al. High dose vitamin E attenuates diabetic nephropathy via alleviation of autophagic stress. Front Physiol 2018; 9:1939.

Treatment with vitamin E restore activities of antioxidant enzymes, probably due to increased Nrf‐2 transcription factor expression and inhibiting lipid peroxidation chain reactions [Bozaykut et al., 2014], suggesting a possible renoprotective effect.

Bozaykut. P.; Karademir. B.; Yazgan. B.; Sozen. E.; Siow. R.C.; Mann. G.E.; Ozer. N.K. Effects of vitamin E on peroxisome proliferator-activated receptor γ and nuclear factor-erythroid 2-related factor 2 in hypercholesterolemia-induced atherosclerosis. Free Radic Biol Med. 2014; 70:174-81.

Our comment on the protection with vitamin E treatment of hypertrophy in proximal renal tubules during diabetes mellitus refers to the observation in studies of histology of the renal cortex and the measurement of the areas of renal tubular epithelial cells, where vitamin E prevents hypertrophy. In subsequent studies we could consider performing the analyzes of RNA expression related to proximal tubule injury markers (Kim1, Krt20, Vcam1, Ccl2, and Cd44) that you mention. As well as check fibrosis marker genes expression (fibronectin 1, transforming growth factor beta 1 and collagen 1A1. TGF beta 1 has been already quantified it in the renal cortex of diabetic rats by induction with streptozotocin in other laboratory works (Amato et al., 2016). Amato, D. Núñez-Ortiz, A. Carmen Benítez-Flores, J. Segura-Cobos, D. López-Sánchez, P. and Vázquez-Cruz, B. (2016) Role of Angiotensin-(1-7) on Renal Hypertrophy in Streptozotocin-Induced Diabetes Mellitus. Pharmacology & Pharmacy7, 379-395. doi: 10.4236/pp.2016.79046.

  1. They showed AT1R and AT2R protein expression in renal cortex homogenate, but they should show the data from one photo, because they combined the different photos to compare the protein expression level.

For the western blot analysis of the proteins of the AT1 and AT2 receptors, as well as the reference protein beta actin, different exposures to the corresponding primary antibodies are carried out, hence different photographic developments are carried out, in addition to sufficient electrophoretic runs to observe the presence of the different proteins that interest us in all samples or experimental repetitions.

Round 2

Reviewer 2 Report

Comments and Suggestions for Authors

As I wrote, in this study, they should examine RNA expression related to proximal tubule injury markers (Kim1, Krt20, Vcam1, Ccl2, and Cd44) to suggest that vitE supplementation protects the proximal tubules and to compare with similar paper on vit E by other groups.

Author Response

Revisor 2

February 7, 2025

Regarding the observation issued by review 2 regarding the usefulness of quantifying the effect of treatment with vitamin E in the diabetic nephropathy model on indicators of damage in proximal convoluted tubules (Kim1, Krt20, Vcam1, Ccl2, and Cd44), it seems relevant to us to do it, although shortly we would not be able to do it as we do not have the required experimental elements. Reviewing the scientific literature on the matter, we did not find works that had already studied it. We found the analysis of these indicators of kidney damage in other models of nephrotoxicity, which we discuss below.

Diclofenac (30 mg/kg, orally, 10 days) is nephrotoxic and caused a significant increase in serum urea, creatinine, KIM-1, TNF-α, NF-κB, and malondialdehyde levels compared to the rats control group. In the rat groups treated with diclofenac plus vitamin E (250 mg/kg, orally), a significant reduction (P<0.05) in the levels of pro-inflammatory cytokines and malondialdehyde was observed, along with improvement in renal function indices, superoxide dismutase, catalase, and glutathione peroxidase levels comparable to the control group. The observed renal histopathological changes were consistent with the results of the biochemical parameters between the treated groups and the normal control rats (Dada et al., 2023).

Dada S, Fabiyi-Edebor T, Akintoye O, Ezekpo O, Dada O, Bamikefa T, Sanya J. α-Tocopherol Ameliorates Redox Equilibrium Disorders and Reduces Inflammatory Response Caused by Diclofenac-Induced Nephrotoxicity in Male Wistar Rats. Cureus. 2023 13;15(12):e50474. doi: 10.7759/cureus.50474. 

10 weeks of supplementation with 600 IU alpha-tocopherol in hemodialysis patients reduced vascular inflammatory markers ICAM-1 and VCAM-1 levels (Pirhadi et al., 2020).

Pirhadi-Tavandashti N, Imani H, Ebrahimpour-Koujan S, Samavat S, Hakemi MS. The effect of vitamin E supplementation on biomarkers of endothelial function and inflammation among hemodialysis patients: A double-blinded randomized clinical trial. Complement Ther Med. 2020;49:102357. doi: 10.1016/j.ctim.2020.102357.

Round 3

Reviewer 2 Report

Comments and Suggestions for Authors

Nothing